# RAC1B: A Rho GTPase with Versatile Functions in Malignant Transformation and Tumor Progression

**DOI:** 10.3390/cells8010021

**Published:** 2019-01-04

**Authors:** Catharina Melzer, Ralf Hass, Hendrik Lehnert, Hendrik Ungefroren

**Affiliations:** 1Biochemistry and Tumor Biology Lab, Department of Obstetrics and Gynecology, Hannover Medical School, 30625 Hannover, Germany; Melzer.catharina@mh-hannover.de (C.M.); hass.ralf@mh-hannover.de (R.H.); 2First Department of Medicine, UKSH, Campus Lübeck, 23538 Lübeck, Germany; hendrik.lehnert@uni-luebeck.de; 3Department of General and Thoracic Surgery, UKSH, Campus Kiel, 24105 Kiel, Germany

**Keywords:** RAC1B, RAC1, cancer, Rho GTPase, proliferation, survival, signaling, epithelial-mesenchymal transition

## Abstract

RAC1B is an alternatively spliced isoform of the monomeric GTPase RAC1. It differs from RAC1 by a 19 amino acid in frame insertion, termed exon 3b, resulting in an accelerated GDP/GTP-exchange and an impaired GTP-hydrolysis. Although RAC1B has been ascribed several protumorigenic functions such as cell cycle progression and apoptosis resistance, its role in malignant transformation, and other functions driving tumor progression like epithelial-mesenchymal transition, migration/invasion and metastasis are less clear. Insertion of exon 3b endows RAC1B with specific biochemical properties that, when compared to RAC1, encompass both loss-of-functions and gain-of-functions with respect to the type of upstream activators, downstream targets, and binding partners. In its extreme, this may result in RAC1B and RAC1 acting in an antagonistic fashion in regulating a specific cellular response with RAC1B behaving as an endogenous inhibitor of RAC1. In this review, we strive to provide the reader with a comprehensive overview, rather than critical discussions, on various aspects of RAC1B biology in eukaryotic cells.

## 1. Introduction

RAC1B is a member of the Rho family of small GTPases and, like RAC1, is a product of the *RAC1* gene. The majority of data on this RAC-related isoform have been gained from tumor cell models and these strongly support a role of RAC1B in cancer as well as in biological processes that either predispose to cancer like chronic inflammation or initiate its early development. The aim of this review is to serve as a comprehensive manual allowing the interested reader to quickly look up specific aspects of RAC1B biochemistry, cellular functions, signaling interactions, and pharmacological targeting. Finally, we summarize available evidence for its emerging role as a prognostic marker in specific tumor entities.

## 2. RAC1B in the Evolution of Ras-like GTPases

To reveal the evolutionary history of the Rho family of small GTPases, Boureux and colleagues have analyzed over 20 species covering major eukaryotic clades from unicellular organisms to mammals, and have reconstructed the ontogeny and the chronology of emergence of the different subfamilies [1]. The 20 mammalian Rho members fall into 8 subfamilies, with Rac (a common ancestor of RAC1, -2, and -3) being the founder of the entire family. The Cdc42, Rho, RhoBTB and RhoUV subfamilies are the most ancient ones as they emerged before Coelomates while RhoDF, RhoJQ, and Rnd first appeared in chordates. Interestingly, RAC1B emerged in amniotes and RhoD only in therians and thus were the latest members to arise [1].

## 3. General Structure and Tissue Expression of RAC1B

*RAC1* but not *RAC2* or *RAC3* contains an additional exon 3b that is included by alternative splicing into the variant RAC1B, hence *RAC1* encodes two signaling GTPases [2]. The exon 3b of *RAC1* contains additional 57 nucleotides and this results in an in-frame insertion of 19 new amino acids between codons 75 and 76 of *RAC1* immediately behind the switch II region, including two potential threonine phosphorylation sites for casein kinase II and protein kinase C. This *RAC1* splice variant, RAC1B, was predominantly identified in skin and epithelial tissues from the intestinal tract [2] and in breast tissues [3].

## 4. Biochemical Properties, Generation and Degradation of RAC1B

### 4.1. Biochemical Properties

The RAC1B protein acts like a fast cycling GTPase in GTP binding and hydrolysis assays [3]. A structural and biochemical analysis has revealed the structures of RAC1B in the GDP- and the GppNHp-bound forms. They show that the insertion induces an open switch I conformation and a highly mobile switch II. As a consequence, RAC1B exhibits an accelerated guanine nucleotide exchange factor (GEF)-independent GDP/GTP exchange and an impaired GTP hydrolysis, which is restored partially by GTPase-activating proteins (GAPs) [4]. The insertion of exon 3b leads to a reduced affinity for GDP and consequently enhanced intrinsic guanine nucleotide exchange, as well as a decreased intrinsic GTPase activity, resulting the intracellular predominance of the active GTP-bound state of RAC1B. Earlier studies showed that RAC1B exhibited the biochemical features of a constitutively activated GTPase [5]. Thus, RAC1B has similarities to the activated melanoma RAC1-P29S protein with respect to spontaneous activation by substantially increased inherent GDP/GTP nucleotide exchange [6]. RAC1B, however, differs from this RAC1 mutant by the reduced intrinsic GTP hydrolysis which in RAC1-P29S is not affected [6]. The mechanisms of RAC1B and RAC1-P29S activation are thus different from the common oncogenic mutations found in Ras-like GTPases that abrogate GTP hydrolysis [6]. Although the regulation of both RAC1 and RAC1B activities is dependent on GAPs, the difference in their activation is mainly determined by the inability of RAC1B to interact with RHO-GDP dissociation inhibitor (RHO-GDI) [7,8]. As a consequence, most RAC1B remains bound to the plasma membrane and is not sequestered by RHO-GDI in the cytoplasm. Although little RAC1B protein is expressed in cells, the amount of activated RAC1B protein may exceed that of activated RAC1, suggesting that RAC1B contributes significantly to the downstream signaling of RAC. However, the specific biochemical properties of RAC1B have severe consequences for signaling and interaction with downstream effectors that initially led authors to suggest that RAC1B may be defective in biological activity.

### 4.2. Regulation of RAC1B Splicing

As mentioned above, RAC1B is generated from *RAC1* by alternative RNA splicing. Using a *RAC1* minigene in HT29 colorectal cancer (CRC) cells, Gonçalves and coworkers found that the splicing factor SRSF3 (formerly SRp20) increased skipping of alternative exon 3b, whereas another splicing factor, SRSF1 (formerly ASF/SF2), increased its inclusion [9]. RNA interference-mediated knockdown of these splicing factors confirmed that SRSF3 acts as a silencer of endogenous RAC1B splicing, whereas SRSF1 acts as an enhancer. Point mutations in exon 3b defined two adjacent regulatory regions required for skipping or inclusion of exon 3b, which are recognized in vitro by SRSF3 and SRSF1, respectively. Both splicing factors were found to be regulated by upstream signaling pathways. Activation of β-catenin/TCF4 increased expression of SRSF3 and hence inhibited protein levels of RAC1B while the PI3-kinase/AKT pathway increased expression of SRSF1 and promoted RAC1B generation [9].

Alternative splicing of RAC1B via SRSF3 and SRSF1 is affected by Wnt activity in CRC cells. Wnt activity promotes expression of SRSF3, thereby blocking expression of RAC1B. Interestingly, RAC1B itself enhances Wnt signaling, forming a negative feedback loop that controls RAC1B levels through Wnt-mediated SRSF3 expression [10].

In colorectal cells, AKT2, AKT3, GSK3β, and SR protein kinase 1 (SRPK1) increased endogenous expression of RAC1B. Knockdown of AKT2 and AKT3 affected only RAC1B protein levels suggesting a post-splicing effect, while in contrast knockdown of GSK3β or SRPK1 decreased RAC1B alternative splicing through changes in SRSF1 [11].

Pelisch and colleagues investigated the molecular mechanisms by which stromelysin-1/matrix metalloproteinase-3 (MMP3), a stromal enzyme upregulated in many breast tumors and other stroma-rich cancers, leads to induction of RAC1B expression. Prompted by the observation that in normal breast tissue and breast cancer biopsies, expression of the splicing factor heterogeneous nuclear ribonucleoprotein (hnRNP) A1 and RAC1B was inversely correlated, they proposed the existence of regulatory interactions between these proteins. HnRNP A1 binds to *Rac1* exon 3b in mouse mammary epithelial cells and represses its inclusion into mature mRNA. Exposure of cells to MMP3 leads to release of hnRNP A1 from exon 3b and the consequent generation of Rac1b [12]. Importantly, hnRNP A1 (and SPSB1) have a critical role in epidermal growth factor (EGF)-driven cell migration. Mechanistically, EGF-induced ubiquitination of hnRNP A1 together with the activation of SRPKs results in the upregulation of RAC1B to promote cell motility in HeLa cells [13].

Induction of RAC1B can also be induced by knockdown of another splicing factor, ESRP1. Downregulation of ESRP1 is closely associated with a motile phenotype of head and neck carcinoma cells. Hence, inhibition of RAC1B generation may be a potential mechanism through which ESRP1 can suppress cancer cell motility [14].

### 4.3. Regulation of RAC1B Protein Stability

Although proteasomal activity is discussed for RAC1B turnover [15], the poor ubiquitination suggests reduced proteolysis in contrast to RAC1-Q61L, a constitutively active RAC1 mutant, which undergoes polyubiquitination and subsequent proteasomal degradation in HEK293 cells. This feature of RAC1B may be of importance for the expression and tumorigenic capacity of RAC1B. Mutational analysis of all lysine residues in RAC1 revealed that the major target site for RAC1 ubiquitination is Lys147, a solvent-accessible residue that has a similar conformation in RAC1B. Like RAC1-Q61L, RAC1B was found to be largely associated with the plasma membrane, a known prerequisite for RAC1 ubiquitination. Ubiquitination of activated RAC1 occurs through a jun-N-terminal kinase (JNK)-activated process. The defective ubiquitination of RAC1B may thus be explained by RAC1B’s inability to activate JNK (see Section 6.1). Interestingly, RAC1B ubiquitination could be stimulated by coexpression of RAC1-Q61L [16], suggesting positive regulation of RAC1B ubiquitination by RAC1 downstream signaling.

Enhanced stability of RAC1B protein is under negative control by nuclear Wnt signaling and RAC1B levels are regulated by being targeted for degradation through a negative feedback loop initiated by Wnt signaling. This is possible because RAC1B itself can activate the Wnt pathway [15] (see Section 6.4).

### 4.4. Regulation of RAC1B Biological Activity by Subcellular Localization

RAC1B biological activity is strongly regulated by distribution in distinct subcellular compartments. Thus, membrane (and nuclear) localization promotes activity whereas cytoplasmic localization inhibits it. Most RAC1B protein is usually bound to the plasma membrane and is not sequestered by RHO-GDI in the cytoplasm which is a consequence of the inability of RAC1B to interact with RHO-GDI [7,8]. Membrane localization of RAC1B also appears to be dependent on the cellular microenvironment and requires that the cells are cultured on stiff substrata or in collagen-rich regions as they are found in stroma-rich tumors [17]. At the membrane, RAC1B forms a complex with NADPH oxidase and promotes the production of reactive oxygen species (ROS), expression of Snail, and activation of epithelial-mesenchymal transition (EMT). In contrast, soft microenvironments inhibit the membrane localization of RAC1B and subsequent redox changes [17]. Stiff substrata are also a prerequisite for MMP3 induction of RAC1B. In rigid microenvironments, MMP3 upregulates expression of RAC1B, which translocates to the cell membrane to promote induction of ROS and EMT. While α5-integrin maintains RAC1B at the membrane and is required for the promotion of EMT by fibronectin, α6-integrin sequesters RAC1B from the membrane and is required for inhibition of EMT by laminin [18]. Interestingly, the subcellular distribution of RAC1B can have prognostic value with respect to patient outcome [19] (see Section 7.5).

RAC1B, like RAC1, possesses a polybasic region that contains a canonical nuclear localization signal (NLS), (K(K/R)X(K/R)), responsible for nuclear localization. Its mutation resulted in loss of RAC1B stimulatory effects on transcription and suppressive effects on adhesion, indicating, in addition to membrane localization, the importance of nuclear localization of RAC1B [20].

### 4.5. Binding Partners and Downstream Effectors

As mentioned above, RAC1B is unable to interact with RHO-GDI [7,8], resulting in most RAC1B protein being bound to the plasma membrane and not sequestered by RHO-GDI in the cytoplasm. Interestingly, RAC1B is able to bind the GTPase-binding domain of p21-activated kinase (PAK) but not full-length PAK [8] in a GTP-dependent manner. When compared with RAC1 and RAC1-GV12, RAC1B binds with much lower affinity or not at all to many common effectors of Rho GTPases, e.g., RHO-GDI, GIT-1, and IQGAP, but it does display enhanced binding to proteins involved in transcriptional regulation, cell-cell adhesion, and motility, such as SmgGDS, an atypical GEF with multiple armadillo repeats, RACK1 (receptor for activated C kinase 1), a scaffolding protein involved in key signaling pathways, and p120(ctn) [21]. The interaction between RAC1B and p120(ctn) is crucial for RAC1B-induced chemotaxis and is dependent upon protein regions predicted to be unstructured in the absence of molecular complex formation. This suggests that the interaction between RAC1B and p120(ctn) involves coupled folding and binding [21]. Finally, in hepatocellular carcinoma (HCC), the GAP Rho GTPase-activating protein 11A (ARHGAP11A) has been shown by coimmunoprecipitation assay to directly interact with RAC1B independent of Rho GTPase-activating activity and to promote the HCC malignant phenotype [22].

## 5. Biological Functions of RAC1B

### 5.1. Cancer Progression/Cellular Transformation

A series of studies suggests that RAC1B contributes to cellular transformation. Rac1b, like the constitutively-activated and transforming Rac1-Q61L mutant, promoted cellular transformation, e.g., a loss of density- and anchorage-dependent growth inhibition of NIH3T3 mouse fibroblasts [7]. RAC1B overexpression may also facilitate tumor progression by interaction with certain tumor-associated signaling pathways (either functional or deregulated as a consequence of oncogene activation or tumor suppressor protein inactivation). The expression of RAC1B was found to be elevated in CRC at various stages of neoplastic progression. In this tumor type, RAC1B can enhance Dishevelled-3 (DVL3)-mediated Wnt pathway signaling and induction of Wnt target genes specifically involved in decreasing the adhesive properties of CRC cells [20]. RAC1B is also overexpressed most prominently in stages 1 and 2 of human lung adenocarcinomas [23]. In a mouse model of this cancer type, in which the expression of Rac1b can be conditionally activated in the lung, expression of Rac1b alone was insufficient to initiate lung tumor formation but cooperation with oncogenic K-Ras resulted in enhanced proliferation and accelerated tumor growth [24]. Other potential protumorigenic mechanisms include the ability of RAC1B to counteract oncogene (B-RAF-V600E)-induced senescence (OIS) in CRC cells, indicating the selection for increased RAC1B expression as a potential mechanism by which these cells can escape from OIS (see Section 5.8). In addition, MMP-induced expression of Rac1b gave rise to lung adenocarcinoma in transgenic mice through bypassing OIS [23]. Very recently, it was reported that ARHGAP11A facilitates malignant progression in HCC patients via ARHGAP11A-RAC1B interaction [22].

Kotelevets and coworkers induced ectopic expression of Rac1b in mouse intestinal epithelial cells after crossing Rosa26-LSL-Rac1b and villin-Cre mice [25]. These animals were bred with Apc^Min/+^ or IL10^−/−^ mice to trigger intestinal tumors. Rac1b ectopic expression increased intestinal epithelial cell proliferation and migration and enhanced the production of ROS. Although Rac1b overexpression alone was not sufficient to drive intestinal neoplasia, it enhanced Apc-dependent intestinal tumorigenesis. In IL10 knockout mice, the Rac1b transgene promoted cecum and proximal colon carcinogenesis. In contrast, when the above mice received treatment with azoxymethane (AOM)/dextran sulfate sodium (DSS), Rac1b alleviated carcinogen/acute inflammation-associated colon carcinogenesis resulting in part from the early mucosal repair after resolution of inflammation. These data highlight the critical role of Rac1b in cooperating with Wnt pathway dysregulation and chronic inflammation to promote intestinal carcinogenesis [25].

RAC1B has protumorigenic function also in thyroid carcinoma. NFκB activation has been implicated as one of the molecular mechanisms associated with the protumorigenic advantage of RAC1B overexpression in thyroid carcinomas [26]. Figure 1 provides a summary on the protumorigenic mechanisms discussed above.

### 5.2. Inflammation and Regeneration

It is well established that chronic inflammation can predispose to cancer. Kotelevets and colleagues have studied the role of Rac1b in this process using the same mouse model as above (Rosa26-LSL-Rac1b × villin-Cre) but bred with IL10^−/−^ mice after challenge with DSS to induce experimental colitis. In an IL10 knockout background, the Rac1b transgene enhanced colonic inflammation due to induced intestinal mucosa permeability resulting in cecum and proximal colon carcinogenesis (see Section 5.1). However, Rac1b also promoted early mucosal repair after resolution of inflammation which counteracts carcinogen/acute inflammation-associated colon carcinogenesis. These data emphasize the crucial role of Rac1b in driving wound-healing after resolution of intestinal inflammation [25]. Although in this experimental system ectopic expression of Rac1b does not mimic the physiological levels of endogenous Rac1b, endogenous expression of this protein is indeed increased in patients with an inflamed colonic mucosa as well as in mice following experimentally induced colitis. This suggests that inflammation can trigger changes in RAC1B expression in the colon [27].

### 5.3. Stromal Control of RAC1B by MMP3

Rac1b has been shown to be induced by components of the tumor stroma such as MMP3. Exposure of murine mammary epithelial cells to MMP3 induces the expression of Rac1b, which translocates to the cell membrane to promote an increase in cellular ROS which in turn stimulate the expression of Snail. Thereby, MMP3 can cause EMT, oxidative DNA damage, genomic instability, and malignant transformation in cultured cells, and in genomically unstable mammary carcinomas in transgenic mice [28]. MMP3-induced upregulation of Rac1b in mouse mammary epithelial cells requires stiff substrata/rigid microenvironments with compliances characteristic of breast and other stroma-rich cancers [17]. Similarly, exposure of PDAC-derived cells to recombinant MMP3 stimulates expression of RAC1B, increases cellular invasiveness, and activation of tumorigenic transcriptional profiles [19]. Moreover, expression of mouse Rac1b in bitransgenic mice (with inducible expression of Rac1b only in lung epithelial cells) stimulated EMT and spontaneous tumor development. This activation of EMT by MMP3-induced expression of Rac1b gave rise to lung adenocarcinoma in the transgenic mice through bypassing OIS [23].

### 5.4. Epithelial-Mesenchymal Transition (EMT)

Primary steps of metastases formation include cell detachment from the original primary tumor which in many cases involves EMT although previous work has demonstrated that EMT is not necessarily required for all kinds of metastatic behavior [29,30].

As outlined above, RAC1B is a key mediator of MMP3-induced EMT in breast, lung, and pancreas carcinomas (Figure 2). The first evidence that RAC1B can promote EMT came from a study upon exposure of cultured murine mammary epithelial cells to MMP3 which induces the expression of Rac1b and causes an increase in cellular ROS and ROS-dependent stimulation of the expression of the EMT-associated transcription factor Snail [28,31]. MMP3/Rac1b-mediated EMT requires cell spreading to cover a larger surface, a process which is induced in cells by treatment with MMP3 or in cells with ectopic expression of Rac1b [32]. The EMT phenotype is not stable but is shaped by environmental cues and is promoted by stiff/rigid substrata, with compliances characteristic of breast tumors, while soft substrata, with compliances comparable to that of normal mammary tissue, are protective against EMT. Strikingly, in cells cultured on stiff substrata or in collagen-rich regions of human breast tumors, Rac1b localizes to the plasma membrane where it complexes with NADPH oxidase and promotes the production of ROS, expression of Snail, and activation of the EMT program. In contrast, soft microenvironments inhibit the membrane localization of Rac1b and subsequent redox changes. Hence, Rac1b represents a crucial component of a mechanotransduction pathway regulating epithelial plasticity [17]. Also in mammary epithelial cells sequestering of Rac1b from the membrane by α6-integrin is required for inhibition of EMT by laminin. In contrast, α5-integrin-mediated is maintenance of Rac1b at the membrane is required for promotion of EMT by fibronectin [18]. Moreover, interference of cell contact-regulated communication by MMP3 and further cooperation with activated Rac1b is involved in EMT and plays a key role in early stages of breast cancer development. Specifically, treatment of cells with MMP3 via Rac1b interrupts a defined subset of cell contact-regulated genes, including genes that encode RNA splicing proteins known to regulate the expression of Rac1b [33]. A RAC1B-mediated release of ROS is also elicited in prostate carcinoma cells in response to metalloproteases secreted by cancer-associated fibroblasts (CAFs) [34]. RAC1B also seems to regulate the expression of other EMT markers in addition to Snail. RNAi-mediated suppression of RAC1B resulted in a concomitant increase in E-cadherin transcript levels [15], suggesting that RAC1B negatively regulates E-cadherin expression and positively controls expression of Snail2/Slug, a specific transcriptional repressor of E-cadherin. Interestingly, ARHGAP11A utilizes RAC1B to facilitate EMT in HCC cells while upregulation of RAC1B was capable of protecting cells from undergoing the reverse process, mesenchymal-to-epithelial transition (MET) that resulted from ARHGAP11A knockdown [22].

The studies cited so far focused on the role of RAC1B in MMP-induced EMT. However, a recent report investigated EMT induced by TGF-β1 in PDAC-derived cells. Intriguingly, the authors observed an inhibitory role of RAC1B (Figure 2). RNAi-mediated, exon 3b-targeted knockdown of RAC1B increased an EMT phenotype as evidenced by cell morphology, gene expression of EMT markers, cell migration (see Section 5.6), and growth inhibition [35]. The suppressive effect of RAC1B on EMT was further substantiated by the observation that the activities of SMAD and non-SMAD (MKK3/6-p38 MAPK and MEK-ERK1/2) signaling pathways, all of which are involved in TGF-β-induced EMT in pancreatic cells, were increased upon RAC1B knockdown [35].

The role of RAC1B in EMT appears thus inconsistent in that it can be both an inducer or an inhibitor of EMT (Figure 2).

### 5.5. Adhesion and Cell-Cell Interactions

RAC1B overexpression has been shown to decrease the adhesive properties of CRC cells by enhancing DVL3-mediated Wnt pathway signaling and induction of Wnt target genes specifically involved in decreasing the adhesive properties of CRC cells [20]. As mentioned above (see Section 4.4), this ability of RAC1B was dependent on a functional NLS and hence its nuclear localization.

RAC1B activation controls cell-cell interactions, i.e., the assembly of the adherens junctions by mediating laminin-5-integrin-induced signaling by PI3-kinase in HT29 cells [36]. However, in contrast to constitutively active RAC1, RAC1B was unable to induce the disassembly of E-cadherin complexes from junctions in human keratinocytes [37], which may be due to the inability of RAC1B to bind and activate PAK [8].

### 5.6. Cell Motility and Migration

Indirect evidence that RAC1B is involved in regulating cell motility already came from the EMT-promoting role of RAC1B (see Section 5.4) and from a study showing that downregulation of the splicing factor ESRP1 is closely associated with a motile phenotype of cancer cells and induction of RAC1B [14]. More direct evidence was first provided by a study showing that directed cell movement initiated by RAC1B is dependent upon p120 catenin (p120(ctn)), an armadillo domain protein involved in multiple cellular functions [21]. As mentioned earlier, EGF-induced ubiquitination of hnRNP A1 together with the activation of SRPKs results in the upregulation of RAC1B to promote cell motility [13]. Indirect regulation of RAC1B alternative splicing may underlie the influence of the neural crest-associated transcription factor FOXD1 on invasion and migration. Knockdown of RAC1B significantly impaired cell migration and invasion capacities of multipotent neural crest cells and metastatic melanomas [38]. Recently, it was shown that RAC1B can mediate ARHGAP11A-induced migration and invasion of HCC cells in vitro and metastasis formation in vivo [22].

While the majority of studies suggested RAC1B to be a promigratory factor, RAC1B may also be able to act as a migration inhibitor, e.g., in cell motility driven by TGF-β. Data obtained from different cell lines including pancreatic ductal epithelial cells and highly metastatic human breast carcinoma cells revealed that Rac1 and RAC1B can significantly modulate TGF-β1-induced cell migration. This effect is associated with antagonistic functionalities whereby RAC1 promotes and RAC1B inhibits TGF-β1-dependent random cell migration (chemokinesis) [39,40]. Data also suggest that TGF-β1-dependent chemokinesis may be coupled to a senescent phenotype in normal breast epithelial cells [39].

Cell migration requires the formation of lamellipodia, a process which is mainly controlled by RAC1. Whether RAC1B, too, can induce the formation of lamellipodia remains contradictory. Shortly after the discovery of RAC1B it was reported that RAC1B induces the formation of lamellipodia in fibroblasts [2]. The same group later found that expression of neither wild-type RAC1B nor constitutively active RAC1B-Q61L led to significant induction of lamellipodia in SW480 human colon cancer cells [8]. Whether cell type-specific differences account for these discrepant observations needs further clarification.

### 5.7. Proliferation/Cell Cycle Regulation

One of the best studied responses to RAC1B activation in tumor cells is proliferation and cell cycle progression. In mouse fibroblasts expression of wild-type RAC1B was sufficient to stimulate CYCLIN D1 accumulation and G1/S progression, and both responses were blocked by the NFκB super-repressor IκBα(A32A36) [41]. Similarly, in colorectal cells depletion of RAC1B by siRNAs was associated with a reduced G1/S progression rate and inhibited endogenous NFκB activation [42]. In another cell model, HEK293T cells stably overexpressing RAC1 or RAC1B, the level of CYCLIN D1 was significantly increased in both serum-starved RAC1 and RAC1B overexpressing cells. RAC1B overexpression also stimulated G1/S progression and induced a significant increase in *CCND1* reporter activity in the K1 PTC-derived thyroid cell line [25]. In HEK293T and SW480 cells stably overexpressing RAC1B, the pro-proliferation genes *MAPK9* (encoding JNK2), *JUN* and *CCND1* were all upregulated and RAC1B promoted cell proliferation by activating the JNK2-c-JUN-CYCLIN D1 pathway [43] (Figure 3). RAC1B was also required for neurotrophin (NT3)-stimulated cell proliferation of human mesenchymal stem cells (hMSCs) [44].

Furthermore, RAC1B may synergize with oncogenes to stimulate proliferation. In primary colorectal tumors and colorectal cell lines RAC1B overexpression and B-RAF-V600E are significantly associated and the simultaneous suppression of both proteins dramatically impaired cell cycle progression [45]. Moreover, synergism of RAC1B with K-Ras-G12V in a mouse model of lung adenocarcinoma enhanced proliferation and tumor growth [24]. Finally, RAC1B may indirectly stimulate proliferation by blocking the action of growth inhibitory factors. Ungefroren and coworkers have found that RAC1B acts to block growth inhibition by TGF-β1 in pancreatic [35] and breast cancer cells [46]. RAC1B knockdown resulted in upregulation of the cyclin-dependent kinase (cdk) inhibitor p21^WAF1^ [35], a known mediator of TGF-β-induced growth arrest. Whether RAC1B also acts as a TGF-β signaling inhibitor in normal/benign precursor cells, thereby compromising the tumor suppressor activity of TGF-β and eventually promoting malignant transformation (Figure 3) remains an interesting issue for future research.

While the majority of studies reported a proliferation-promoting role of RAC1B under conditions of ectopic overexpression, others made observations that do not support such a role. For instance, in NIH3T3 cells, Rac1b, unlike activated Rac1-Q61L, did not stimulate transcription from the *CCND1* promoter [7]. Rac1b was also not required for K-Ras-driven cell proliferation [24]. In Madin-Darby Canine Kidney cysts, a model for polarized epithelial structure, expression of Rac1b, unlike wild-type Rac1 or Rac1-G12V, did not promote cell cycle progression, leading the authors to conclude that the expression of Rac1b per se cannot induce cell proliferation [47].

### 5.8. Cellular Senescence

Interestingly, it was shown that with the help of RAC1B, cells may be able to overcome OIS. Expression of mouse Rac1b in lung epithelial cells of transgenic mice stimulated EMT and spontaneous tumor development. Through bypassing OIS, MMP-induced Rac1b was able to induce lung adenocarcinoma in the transgenic mice [23].

Oncogenic B-RAF-V600E expressed in normal NCM460 colonocytes induced a senescent phenotype as evidenced by expression of senescence-associated β-galactosidase (SA-β-gal), the cell cycle inhibitors p14^ARF^, p15^INK4B^ and p21^WAF1^, and decreased proliferation. Upon coexpression of RAC1B, but not RAC1, the B-RAF-induced senescence phenotype was reverted and expression of the cell cycle inhibitors downregulated in a ROS-dependent manner. Hence, coexpression of RAC1B counteracts B-RAF-induced senescence, indicating increased RAC1B expression as one potential mechanism by which colorectal tumor cells can escape from OIS [48] (Figure 3).

In pancreatic cancer cells [35] and normal human mammary epithelial cells (HMEC) [46], RAC1B acts as a negative regulator of the TGF-β1-induced cell cycle inhibitor p21^WAF1^ which supports proliferative effects of RAC1B on cell cycle progression (see Section 5.7). More evidence for this was obtained using an aging model of HMEC. A majority of HMEC cultures can spontaneously exit the proliferative cell cycle and enter a senescent growth-arrested phenotype after 15-16 passages (P15/P16) in vitro as demonstrated by upregulation of SA-β-gal and the tumor suppressor and cdk inhibitor p16^INK4A^ [49,50,51]. Autocrine production of TGF-β enhanced the senescent phenotype of HMEC whereby premature senescence is often associated with classical TGF-β-mediated responses like growth inhibition, upregulation of p16^INK4A^, p21^WAF1^, and production of extracellular matrix (ECM) components [51]. Indeed, senescent HMEC display enhanced activity of TGF-β signaling which is supported by TGF-β-mediated pathways to confer upregulation of SMAD2 and SMAD4, p16^INK4A^ [52], COL4A2 [53], MMP2 [54], and uPAR [55]. The induction of these genes in senescent HMEC is paralleled by the observation that an accumulation of aging HMEC is associated with enhanced matrix production [56], a hallmark response to TGF-β. In accordance with the promigratory role of TGF-β on mammary epithelial cells, senescent HMEC in P15/P16 but not their juvenile counterparts in P11/P12 exhibit strongly enhanced chemokinesis [46]. These findings are paralleled by lower expression of RAC1B and higher expression of RAC1 in the motile high-passage senescent HMEC [46], substantiating inverse regulatory interactions between TGF-β signaling and RAC1B expression. Consequently, RAC1B appears to function as an inhibitor of both TGF-β-dependent migration and TGF-β-induced growth arrest with subsequent induction of senescence. While TGF-β can relay Ras-mediated senescence in HMEC thus preventing malignant transformation [57], it appears conceivable that elevated RAC1B expression allows for an escape from tumor-suppressive senescence and progression towards neoplastic development even in the absence of oncogene activation. Together, the senescence-like growth arrest and cell migration in mammary epithelial cells may be controlled at least in part by RAC1B through its inhibitory effect on the RAS-RAF-MEK-ERK pathway [35] and the TGF-β/SMAD pathway [40] (Figure 3).

### 5.9. Survival/Anti-Apoptosis

Expression of wild-type Rac1b, but not of wild-type Rac1, in NIH3T3 cells dramatically increased cell survival in the presence of only minimal growth stimuli [41]. Colorectal cells also depend on RAC1B signaling to sustain their survival. Depletion of RAC1B by siRNA reduced cell viability which was due to increased apoptosis [42]. RAC1B overexpression protected thyroid cells (the K1 PTC derived cell line) from undergoing apoptosis [26]. In all three cell types, the prosurvival effect of RAC1B was dependent on activation of canonical NFκB signaling as evidenced by p65 nuclear localization, an increase in NFκB reporter activity, a decrease in protein levels of the NFkB inhibitor IκBα, and the ability of the NFκB super-repressor IκBα(A32A36) to block this cellular response. Ying and coworkers observed that the survival of HEK293T cells stably overexpressing RAC1B was significantly increased in serum-starved media compared to HEK293T cells stably overexpressing RAC1 in which survival was only moderately increased [58]. Also in HEK293T cells (and in SW480 cells) stably overexpressing RAC1B, the anti-apoptotic proteins AKT2 and MCL1 were upregulated and the AKT2/MCL1 pathway activated coincident with inhibition of apoptosis. Conversely, knockout of the *RAC1* exon 3b or knockdown of endogenous RAC1B in HT29 cells downregulated the AKT2/MCL1 pathway [43]. Finally, inhibition of RAC1B expression in HT29 cells with the nonsteroidal anti-inflammatory drug ibuprofen (see Section 6.6) reduced cell survival in vitro and RAC1B-dependent tumor growth of HT29 xenografts in vivo [27].

As described above for effects of RAC1B on proliferation, RAC1B can functionally cooperate with B-RAF-V600E in CRC cells to sustain survival and, hence, may form an alternative survival pathway to oncogenic K-RAS [45].

### 5.10. Neurogenic Stem Cell Differentiation

RAC1B can act as a differentiation factor in directing hMSCs (MIAMI) cells toward an early neuronal phenotype. This was accompanied by a negative effect on proneuronal gene expression and neurite-like extensions. RAC1B was also required for NT3-stimulated cell proliferation of MIAMI cells, yet was found to repress CCND1 and CCNB1 mRNA expression independent of NT3 stimulation. These findings demonstrate that the in vitro neurogenic potential of hMSCs is governed by RAC1B during NT3 stimulation [44].

### 5.11. Mutual Negative Regulation of RAC1 and RAC1B

Intriguingly, it was observed that RAC1B negatively regulates RAC1 activity. Specifically, expression of RAC1B in HeLa cells i) blocked RAC1 activation by PDGF, ii) reduced the fraction of membrane-bound RAC1 and iii) promoted an increase in RHO activity. The antagonistic relationship between RAC1 and RAC1B perturbs actin cytoskeleton dynamics and led to alterations in cell morphology and motility [59]. A likely mechanism for this antagonism is competition for activation by GEFs. Along the same lines are findings from our group that exon 3b-targeting siRNA-mediated knockdown of RAC1B in the pancreatic carcinoma cell line Panc1 resulted in upregulation of RAC1 protein levels [35]. Although the mechanistic basis for this is still elusive, it indicates that RAC1B can act to suppress expression of RAC1. A reverse interaction which impacted protein stability has been observed in that degradation of RAC1B via ubiquitination is enhanced upon coexpression of the constitutively active RAC1-Q61L mutant [16] (see Section 4.3). Mutual negative regulation of RAC1 and RAC1B may be involved in one or more of the cellular processes described in
Section 5.4, Section 5.5, Section 5.6, Section 5.7, Section 5.8, Section 5.9 and Section 5.10


## 6. Regulation of Signaling Pathways by RAC1B

RAC1B, like RAC1, affects several cancer-associated signaling pathways (see Figure 4 for an overview).

### 6.1. Mitogen-Activated Protein Kinases (MAPKs)

RAC1B has been reported to control several MAPKs. In hMSCs, RAC1B negatively regulates MEK-ERK signaling during NT3-stimulated neurogenic differentiation and proliferation of these cells [44]. Interestingly, we also observed negative regulation of ERK1/2 activation in pancreatic cancer cells. RAC1B knockdown by siRNA resulted in strongly increased phospho-ERK1/2 levels and sensitized cells towards further upregulation of phospho-ERK1/2 levels by TGF-β1 independent of the kinase function of the TGF-β type I receptor ALK5 [35]. Moreover, RAC1B knockdown also increased the extent and duration of TGF-β1-induced phosphorylation of p38 MAPK in a SMAD4-independent manner [35]. Upregulated expression of RAC1B in lung cancer is significantly associated with sensitivity to the MEK inhibitor PD-0325901, suggesting that RAC1B expression itself is controlled by MEK/ERK signaling [60]. Moreover, RAC1B utilizes activation of the JNK2 pathway to enhance cell survival [43]. However, it has also been reported that unlike RAC1, activated RAC1B is unable to activate JNK in SW480 cells [7] and NIH3T3 cells [8]. Unfortunately, no information on the JNK isoform analyzed (JNK1 or JNK2) is supplied in these publications.

### 6.2. NFκB

Both RAC1 and RAC1B are able to activate NFκB to the same extent [8]. Active RAC1B induced the phosphorylation and membrane translocation of IκBα, a prerequisite for the activation of NFκB [41]. In addition, RAC1 and RAC1B stimulate transcriptional activation from reporter genes driven by NFκB motifs or the *CCND1* promoter in an IκBα- and ROS-dependent manner. However, in colorectal cells RAC1B was unable to induce nuclear translocation of RelB and p52, and to activate transcription from a RelB-specific reporter. These data provide evidence that increased levels of RAC1B in colorectal tumors may promote tumorigenesis by stimulating canonical NFκB signaling while circumventing a negative feedback from the RelB pathway [61]. RAC1B overexpression stimulates NFκB-mediated pro-proliferative and anti-apoptotic signaling also in thyroid cancer cells as judged by an increase in NFκB reporter gene activity and a decrease in IκBα protein levels [26]. However, Singh and colleagues reported in NIH3T3 cells that unlike activated Rac1 (Rac1-Q61L), Rac1b did not exhibit an enhanced ability for transcriptional transactivation of NF*κ*B [7]. In addition, RAC1B, unlike RAC1 signaling, is unable to control the expression of genes regulated by the transcriptional repressor BCL-6, such as the p50 precursor NFκB1/p105 and the cell adhesion molecule CD44 [62].

### 6.3. Reactive Oxygen Species (ROS)

Rac1b utilizes ROS in MMP3-induced EMT, DNA damage and genomic instability [28,31]. Interestingly, CAF-secreted metalloproteases utilize RAC1B and cyclooxygenase 2 (COX2) to induce in carcinoma cells the release of ROS [34]. Of note, in progeria RAC1B is also utilized by ROCK in regulating mitochondrial ROS levels. Activated ROCK phosphorylates RAC1B at Ser71 and increases ROS levels by facilitating the interaction between RAC1B and cytochrome c. Conversely, ROCK inactivation with Y-27632 abolishes the RAC1B-cytochrome c interaction, concomitant with ROS reduction [63].

### 6.4. Wnt/β-Catenin

RAC1B overexpression stimulates TCF-mediated gene transcription, whereas depletion of RAC1B results in decreased expression of the Wnt target gene *CCND1*. RAC1B was capable of functionally interacting with DVL3 but not β-catenin to mediate synergistic induction of Wnt target genes. In agreement, DVL3 but not β-catenin caused increased activation of RAC1B [20]. Follow-up studies have shown that RAC1B resides at the promoters of Wnt target genes, c-Myc and *CCND1*, in HCT116 cells with aberrant Wnt pathway [64]. In HEK293T cells with intact Wnt signaling, RAC1B is tied to these same gene promoters independent of WNT3A stimulation and recruits DVL3 and β-catenin in the absence of WNT3A stimulation, suggesting a novel transcriptional coactivator role of RAC1B in β-catenin/TCF-mediated transcription [64]. By enhancing DVL3-mediated Wnt pathway signaling and induction of Wnt target genes involved in decreasing adhesion of CRC cells, RAC1B overexpression may facilitate tumor progression [64]. RAC1B activation of the Wnt pathway requires the NLS located in the polybasic region (see Section 4.4). Earlier data also suggested that nuclear Wnt signaling negatively regulates RAC1B protein stability (see Section 4.3).

### 6.5. TGF-β/SMAD

RAC1B has been shown to negatively affect the TGF-β signaling pathway in pancreatic cancer cells by suppressing C-terminal phosphorylation, and hence activation, of SMAD2 and SMAD3 by the ALK5 kinase [40]. As a consequence, various TGF-β/SMAD-dependent responses such as EMT and random cell migration are disturbed. Mechanistically, published data [35] together with unpublished data from our laboratory point to a crucial role of the inhibitory SMAD7 in this process.

### 6.6. Inhibitors

Pharmacologic inhibitors are important tools for the dissection of or therapeutic intervention with signaling pathways and are also available for RAC1 and RAC1B.

EHT 1864 was identified to possess high affinity binding to both recombinant RAC1 and recombinant RAC1B, and this association promoted the loss of bound nucleotide, inhibiting both guanine nucleotide association and Tiam1 Rac GEF-stimulated exchange factor activity in vitro. EHT 1864, therefore, arrests RAC1 and RAC1B in an inactive state, preventing its binding to downstream effectors [65].

NSC23766 constitutes a RAC-specific small-molecule inhibitor which effectively inhibits RAC1 binding and activation by Trio or Tiam1 in vitro without interfering with RAC1 interaction with BcrGAP or PAK1. In human prostate cancer PC3 cells, NSC23766 inhibited processes that require endogenous RAC1 activity like proliferation, anchorage-independent growth and invasion [66].

Beausoleil and colleagues tested—in a nucleotide binding competition assay against RAC1 and RAC1B in a cellular Rac GTPase activation assay—a series of berberine, phenantridine and isoquinoline derivatives for their Rho GTPase nucleotide inhibitory activity. The insertion of exon 3b in RAC1B was shown to be sufficient to introduce a conformational change that allow four compounds (#4, #21, #22, #26) to exhibit selective inhibition of RAC1B over RAC1 [67].

As mentioned briefly above, ibuprofen has been shown to inhibit RAC1B expression in HT29 cells and RAC1B-dependent tumor growth of HT29 xenografts. This drug acts through a COX inhibition-independent mechanism by interfering with the alternative splicing event rather than by directly inhibiting RAC1B activity. Ibuprofen has been proposed to represent a highly specific and efficient inhibitor of RAC1B overexpression in colorectal tumors [27].

## 7. RAC1B as a Prognostic Marker in Cancer

### 7.1. Breast Cancer

While high expression levels of RAC1 became evident in neoplastic breast tissue of ductal carcinoma-in-situ, primary breast cancer, and lymph node metastases—with low levels in benign breast tissue, little if any differences in expression levels of RAC1B were reported in benign and malignant breast tissue [3]. In contrast, RAC1B protein along with MMP3 proteins was reported to be both strongly expressed by breast tumor cells and MMP3 gene expression can serve as a prognostic marker for patient survival in breast cancer [68].

### 7.2. Colorectal Cancer (CRC)

RAC1B is overexpressed in some colorectal tumors predominantly in an active GTP-bound state which selectively promotes NFκB activation and signaling [42]. This overexpression of RAC1B in colorectal tumors induces cell cycle progression and cancer cell survival [45], and has been associated with the *BRAF-V600E* mutation. *BRAF* mutations are in turn enriched within the consensus molecular subtype (CMS)1 group of CRC [69]. The CMS classification allows for a categorization of most CRC tumors into one of four subtypes with distinguishing features: CMS1 (MSI Immune), CMS2 (Canonical), CMS3 (Metabolic), and CMS4 (Mesenchymal). CMS1 is characterized by increased expression of genes associated with a diffuse immune infiltrate, strong activation of immune evasion pathways, defective DNA mismatch repair, overexpression of proteins involved in DNA damage repair, and hypermethylation. Patients with CMS1 tumors have very poor survival after relapse, in agreement with worse prognosis of patients with recurring MSI and *BRAF*-mutated CRC tumors [69]. It is conceivable that RAC1B expression, too, is overrepresented in the CMS1 group.

The suitability of RAC1B expression as a predictor of chemotherapy efficacy in metastatic CRC (mCRC) was evaluated by Alonso-Espinaco and colleagues. RAC1B overexpression was a poor survival factor for overall survival and progression-free survival in K-RAS/B-RAF wild-type mCRC patients. However, RAC1B overexpression represents a marker of poor prognosis in K-RAS/B-RAF wild-type mCRC patients treated with first-line FOLFOX/XELOX therapy [70]. Hence, in K-RAS/B-RAF colorectal tumors and corresponding metastases, an overexpression of RAC1B has been suggested as a biomarker of poor prognosis [70].

### 7.3. Hepatocellular Carcinoma (HCC)

ARHGAP11A, which is highly expressed in HCC, promotes a malignant phenotype by facilitating cell proliferation, EMT, invasion and migration in vitro in a RAC1B-dependent manner. This GAP increases total RAC1B protein levels and RAC1B activity in HCC cell lines, although the precise mechanism of this regulation remains unknown at present [22].

### 7.4. Non-Small Cell Lung Cancer (NSCLC)

Compared to healthy controls, RAC1 and RAC1B were found to be significantly overexpressed in the serum of NSCLC patients, independent of the cancer stage. The high specificity and sensitivity obtained from surface plasmon resonance-based quantification of RAC1 and RAC1B qualify these proteins for the use as a diagnostic serum marker in the early stage of NSCLC [71].

### 7.5. Pancreatic Cancer and Chronic Pancreatitis

In pancreatic adenocarcinoma (PDAC), MMP3-mediated RAC1B expression increased cellular invasiveness and tumorigenic potential [19,68]. Interestingly, patient outcome was correlated here with subcellular distribution of RAC1B. Specifically, patients showing apolar RAC1B staining presented with significantly shorter survival than patients showing baseline and polar RAC1B staining [19]. Our own group also identified RAC1B overexpression in the ductal epithelial cells of stage III PDAC tumors and, in addition, in chronic pancreatitis tissues, in which the levels of RAC1B protein even exceeded those in PDAC [40]. Intriguingly, and in contrast to the above study [18], RAC1B expression was higher in long-time vs. short-time survivors among PDAC patients. In the light of in vitro data revealing a role of RAC1B as a TGF-β signaling inhibitor [35,40], we propose that in a protumorigenic TGF-β-rich environment the effects of RAC1B as a tumor suppressor predominate over those as a tumor promoter.

### 7.6. Thyroid Cancer

RAC1B is expressed in thyroid and overexpressed in 46% of papillary thyroid carcinomas (PTCs). RAC1B overexpression was significantly associated with both *BRAF-V600E* mutation and poor clinical outcome. Whereas *BRAF-V600E* alone did not associate with patient outcome, the association of RAC1B overexpression with *BRAF-V600E* was overrepresented in the group with poorer clinical outcome. These data suggest that RAC1B and B-RAF-V600E may interact to worsen the prognosis of PTC patients [72,73].

In follicular thyroid carcinomas (FTCs) RAC1B was found to be overexpressed in 33% of carcinomas while no RAC1B overexpression was documented among follicular adenomas. RAC1B overexpression was significantly associated with both the presence of distant metastases and poorer clinical outcome, suggesting that RAC1B overexpression in FTCs (as in PTCs) is associated with poor outcomes. Furthermore, the lack of RAC1B overexpression in follicular adenomas reveals its potential as a molecular marker for preoperative differential diagnosis of thyroid follicular lesions [72].

## 8. Conclusions and Perspectives

### 8.1. Is RAC1B Really the Bad Brother of RAC1 in Cancer?

The majority of studies available so far demonstrates that RAC1B can promote tumor progression when expressed in a protumoral context, e.g., with activated oncogenes, inactivated tumor suppressor genes, or activated cancer-associated signaling pathways. Its tumorigenic role may be elicited at several levels:by upstream activators that affect its generation and subcellular localization. For instance, PI3-kinase/AKT signaling promotes RAC1B expression via alternative splicing while Wnt signaling inhibits it.by an increase in RAC1B activation levels due to altered regulation by GEFs and GAPs.by functional interactions with factors that promote malignant progression such as oncogenes (BRAF, KRAS) and tumor suppressor genes (APC), or environmental clues such as MMPs.by shifting the balance of pro- to antitumorigenic signaling pathways. For instance, RAC1B activates tumor-promoting NFκB and KRAS signaling but inhibits tumor-suppressive TGF-β signaling.by stimulating cancer-promoting processes such as carcinogen/acute inflammation or protecting against cancer such as early mucosal repair, depending on the tissue.

Depending on the type and physiological state of the target tissue, RAC1B may also be able to counteract malignant progression:by promoting early mucosal healing after resolution of intestinal inflammation.by antagonizing TGF-β signaling under conditions where TGF-β promotes malignant progression, i.e., in advanced stages of many carcinomas.

Despite the predominant evidence for a tumorigenic role, a clear-cut picture on the role of RAC1B in cancer is still lacking. This is mainly because RAC1B does not seem to be tumorigenic per se. For instance, RAC1B overexpression alone was not sufficient to drive intestinal neoplasia or lung cancer progression [24,25]. The controversial role is also reflected by contradictory findings with respect to proliferation control. Several studies were unable to demonstrate effects of RAC1B on cell cycle progression or surrogate markers of mitotic activity (*CCND1* expression/reporter gene activity, BrdU incorporation, Ki67 staining). With respect to EMT, MMP3-induced EMT is promoted by RAC1B, while TGF-β-dependent EMT is inhibited, at least in pancreatic cells. Unfortunately, data on the role of RAC1B in EMT induced by still other agents, e.g., ionizing radiation [74] (see Figure 2), are not available. Likewise, cell migration/invasion is either enhanced or inhibited by RAC1B and whether RAC1B can induce the formation of lamellipodia remains an unresolved issue. These discrepancies may partially be explained by cell or tumor type-specific differences of upstream activators and downstream effectors or by the study design. It should be stressed that many studies were carried out under conditions of ectopic overexpression. Given the usually low protein levels of RAC1B in cells compared to RAC1, ectopically expressed RAC1B may by order of magnitudes exceed those of endogenous RAC1B and it might be speculated that as a result of this unphysiologically high expression the observed effects do not truly reflect those displayed by endogenous RAC1B. In some cellular systems this is further complicated by the ability of RAC1B to inhibit expression and/or activity of RAC1 [35,59]. In cases where both proteins are expressed and act in an antagonistic fashion on the same cellular response, an increase or decrease of RAC1B (regardless of whether physiological or artificial) would not only change the direct effect of RAC1B on this response but would inevitably decrease or increase, respectively, the inverse effect induced by RAC1. Ignoring the phenomenon of RAC1B functioning as an endogenous inhibitor of RAC1 could eventually lead to an overestimation of the RAC1B effect. Finally, several studies have used the different activities of RAC1B vs activating versions of RAC1 to argue that there are differences in the activities of the two splice variants. Usually, RAC1 mutants with defective intrinsic GTPase activity and constitutive activation were used, i.e., RAC1-Q61L. Given the greater similarity of RAC1B with another class of activated RAC1 mutants that are fast-cycling and have at least partially retained the intrinsic GTPase activity, for example RAC1-P29S (see Section 4.1), it would possibly be more conclusive to employ a member of this class rather than a GTP-locked RAC1 mutant for comparison with RAC1B.

Future research should be devoted to the role of RAC1B in cancer, with a particular focus on distinguishing its role from that of RAC1. This is especially important for studies which involved the use of experimental approaches that did not allow for discrimination of the effects of both RAC1 isoforms, i.e., a study using genomic deletion of *Rac1* from pancreatic progenitor cells in different mouse models of PDAC. Deletion of *Rac1* in K-Ras-G12D-induced PDAC in mice reduced formation of acinar-ductal metaplasia (ADM), pancreatic intraepithelial neoplasia (PanIN) and tumors, and significantly prolonged survival, leading the authors to conclude that in mice, Rac1 is required for early metaplastic changes and neoplasia-associated actin rearrangements in development of pancreatic cancer [75]. However, since the genomic deletion knocked out both Rac1 and Rac1b protein, the observed effects could not be unambiguously attributed to Rac1 alone in case of the (likely) coexpression of Rac1b with Rac1 in pancreatic progenitor cells. It would be exciting to compare the effects on ADM, PanIN and tumor formation in these mouse models after selective knockout of either Rac1 or Rac1b.

### 8.2. Potential Prognostic and Therapeutic Use of RAC1B

Despite these unsolved issues, RAC1B may still serve as a prognostic marker and a potential target in several tumor entities in which overexpression has been confirmed [40,68]. Thus, determination of the RAC1/RAC1B balance in a biosensor-based system of real-time analysis by using blood serum of NSCLC patients revealed increasing RAC1 and RAC1B protein levels according to the progressive stage of this disease [71]. In CRC, RAC1B overexpression was revealed as a marker of poor prognosis in K-RAS/B-RAF wild-type mCRC patients treated with first-line FOLFOX/XELOX therapy [70]. Moreover, in PDAC both the expression level in the ductal epithelial/cancer cells as well as the subcellular distribution of RAC1B in these cells can have prognostic value with respect to patient outcome [19,40,68].

Therapeutic targeting of RAC1B to inhibit its expression or to downregulate its activity could represent a promising approach in several types of tumors. Blocking the activity of RAC1B and its associated signaling pathways or shifting the endogenous RAC1B/RAC1 balance toward RAC1 appears a rational treatment of primary tumors with high RAC1B expression such as those of the breast, colon and lung. Even prospectively inhibiting RAC1B expression/activity in the premalignant state, e.g., during intestinal mucosal inflammation, chronic pancreatitis, or in HMEC could be beneficial since it may prevent escape from a TGF-β-induced senescence program with subsequent neoplastic development. Conversely, in advanced stages of PDAC where tumor progression is driven by TGF-β, promoting RAC1B expression or shifting the RAC1B/RAC1 balance towards RAC1B could be feasible [40]. For therapeutic interventions several promising agents have been identified, i.e., four other berberine/phenantridine/isoquinoline derivatives [67], EHT 1864 [65], and ibuprofen [27]. Future efforts should focus on the design of protein–protein interaction inhibitors that could block RAC1B-specific downstream signaling.

## Figures and Tables

**Figure 1 cells-08-00021-f001:**
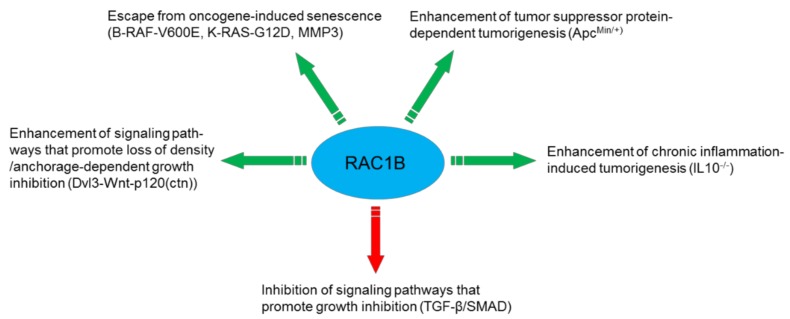
Mechanisms utilized by RAC1B to promote cancer progression.

**Figure 2 cells-08-00021-f002:**
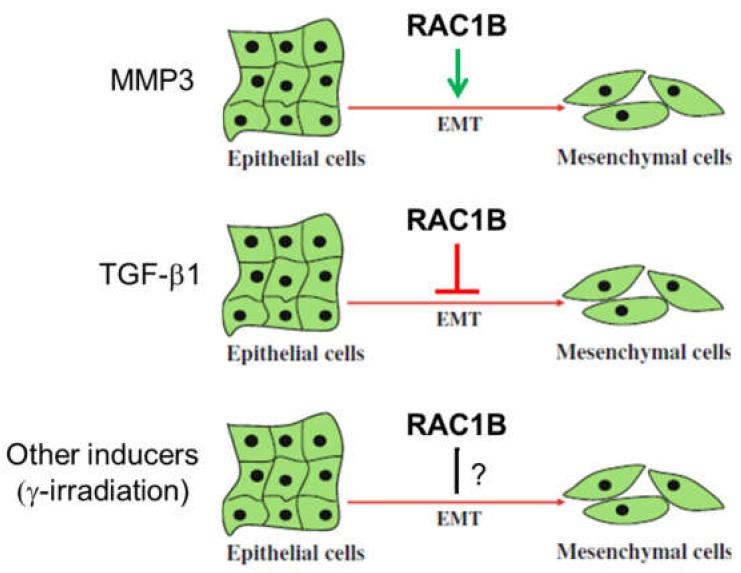
Differential role of RAC1B in epithelial-mesenchymal transition (EMT). Whether RAC1B stimulates or inhibits EMT depends on the EMT-inducing agent. For instance, MMP3 has been reported to promote EMT and TGF-β1 to inhibit it. The role of EMT induced by other mechanisms, e.g., ionizing radiation, still needs to be determined.

**Figure 3 cells-08-00021-f003:**
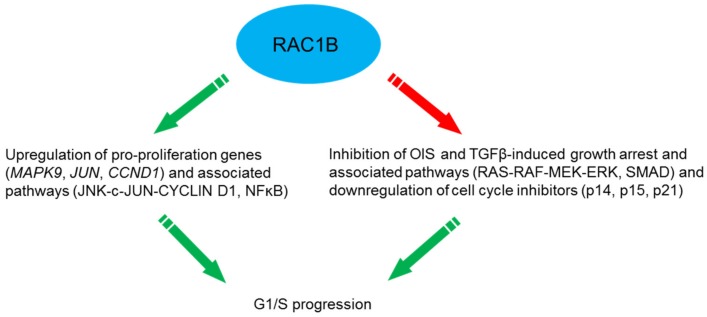
Mechanisms utilized by RAC1B to drive G1/S progression.

**Figure 4 cells-08-00021-f004:**
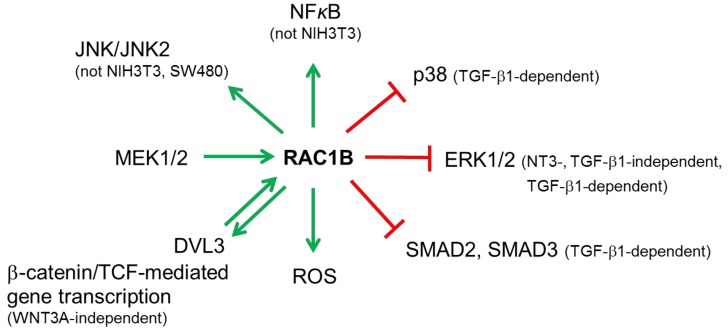
Illustration of the predominant involvement of RAC1B in cancer-associated signaling pathways. Green arrows pointing away from RAC1B indicate activation, those pointing toward RAC1B indicate upregulation of expression (MEK1/2) or activation (DVL3). Red lines indicate suppression. For details see text.

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
