# Peer review of "RAC1B: A Rho GTPase with Versatile Functions in Malignant Transformation and Tumor Progression"

_cells, 2019, doi:10.3390/cells8010021_

Round 1
Reviewer 1 Report
In this manuscript the authors pretended to provide a comprehensive overview on the biology of RAC1b in eukaryotic cells. Given the considerable increase in data published on this splice variant over the last 15 years, this pretension is fully justified.
The authors represent a rather compartmentalized overview, organized in subsections that separately describe gene data, physiological protein properties, biological processes, signaling pathways and finally cancer types. This necessarily leads to some degree of redundant information that is being referred to in various of the above mentioned subsections. However, this organization is useful for the review to be used as a look-up document regarding the roles of RAC1b in various biological aspects and allowed to gather the published information on these aspects that currently is scattered throughout the meanwhile quite numerous publications on RAC1b.
The manuscript contents are well-structured and represent well the state of the art. I have several mostly minor suggestions to improve the manuscript:
1. I recommend to add ‘epithelial-mesenchymal transition’ to the list of keywords
2. The authors may wish to follow the gene name nomenclature guidelines (http://www.genenames.org/about/guidelines#Appendix1) recommending that human gene symbols generally are italicised, with all letters in uppercase (e.g., RAC1), whereas the corresponding protein designations are the same as the gene symbol, but are not italicized (RAC1).
3. paragraph 3, lines 50-59: It is stated that the RAC1 but not the RAC2 gene contains an additional exon; information on the RAC3 gene should be added, too.
4. lines 51-53: The following sentence is ill-constructed: “The exon 3b of RAC1 51 contains additional 57 nucleotides between codons 75 and 76 of RAC1 and immediately behind the 52 switch II region”. The exon 3b itself contains neither codons 75 or 76 nor the switch 2 region. I suggest a fusion with the following phase: The exon 3b of RAC1 contains additional 57 nucleotides and this results in an in-frame insertion of 19 new amino acids between codons 75 and 76 of RAC1 immediately behind the switch II region, including two potential threonine phosphorylation sites for casein kinase II and protein kinase C.
5. The last two sentences of this paragraph (lines 56-58) should be moved to the section 5.1 on cancer progression
6. paragraph 4.1, line 70: The following sentence could be misleading “The insertion of exon 3b leads to enhanced intrinsic guanine nucleotide exchange activity and..”; this is not a real activity that RAC1b has; I suggest to refer only to nucleotide exchange and reduced affinity, for example ‘The insertion of exon 3b leads to a reduced affinity for GDP and consequently enhanced intrinsic guanine nucleotide exchange, as well as a decreased intrinsic GTPase activity, resulting in…
7. lines 73-81: references are missing here that showed the relation between RAC1b and Rho-GDI
8. paragraph 4.2, line 86-94: SR protein family members of splicing factors have been renamed in 2010 (doi: 10.1101/gad.1934910), ASF/SF2 is now SRSF1 and SRp20 is SRSF3.
9. paragraph 5.1: refers to many aspects that are repeated below in other sections, I suggest to focus this paragraph on the two described transgenic mouse models.
10. paragraph 5.2: lines 197: the data described in the first line are not from Matos but Kotelevets and colleagues,
11. paragraph 5: Should mention the mechanistic relation of MMP3 with hnRNP A1 and Rac1b splicing given in ref 10
12. paragraph 5.4: please include in this section the recently published paper on EMT in hepatocarcinoma cells (doi: 10.1186/s12964-018-0312-4) that is cited as reference 65, which is no longer in press and should be updated. I would also suggest to add a comment at the end of this section on the inconsistent observations that RAC1b can be an inducer or an inhibitor of EMT.
13. paragraph 5.5: line 275, please indicate the cell type in which this was observed, in order to distinguish data clearly from those reported for keratinocytes in the same paragraph.
14. paragraph 5.6: Cell motility and migration would be a more appropriate subheading here
15. paragraph 5.10, lines 410-418: The name Rac1b has been used in the past to describe a neuronal-specific chicken Rac gene (Malosio et al., 1997) that was later recognized as homologous to human Rac3 (Albertinazzi et al 2003) (corresponding to references 57 and 58 of the manuscript). The same applies to paragraph 6.6 on ER signalling and ref 64. The corresponding paragraphs should therefore be removed from the manuscript.
16. paragraph 7: The described topics do not justify an independent section. The data on RAC1b interacting proteins (7.1) belong to the end of section 4, following topics on protein stability and subcellular localization.
17. paragraph 7.2 on Inhibitors, should be moved to the end of section 6 on signaling pathways, as inhibitors are important tools for research or pharmacological intervention and their description fits better to this topic than to a section on protein interactions.
18. Regarding the content of paragraph 7.2, both general RAC inhibitors (RAC1 and RAC1b), EHT1864 and NSC23766 should be discussed first before proceeding to the described RAC1b-specific inhibitor data. It would also be useful to distinguish that ibuprofen has been suggested as an inhibitor of the alternative splicing event rather than a direct inhibitor of RAC1b activity.
19. Finally, paragraph 7.3 on mutual negative regulation of RAC1 and RAC1b would fit much better to the end of section 5 that describes the roles on motility, EMT, cell survival etc, which could be modulated by the described mutual negative regulation. In line 555, ‘reduce’ should be ‘reduced’. At the end of line 558; I miss a phrase or expression suggesting that competition for RAC activation by GEFs as a likely mechanism (before proceeding to the remaining paragraph describing possible effects on protein stability).
20. paragraph 8.2: The described association of RAC1b overexpression with the presence of BRAF-V600E should be discussed with respect to the recently proposed Consensus Molecular subtype CMS1 of colorectal cancer (doi:10.1038/nm.3967).
21. paragraph 9.1: This is a very useful and important section that integrates conflicting data form the literature; however, the following paragraph (lines 639-650) needs some clarification considering the given topic” RAC1b’s tumorigenic role can be elicited at several levels:”:
“1. by activity regulation based on the biochemical characteristics.” is unclear. Do you mean ‘ by an increase in RAC1b activation levels due to altered regulation of GEFs and GAPs’??
“3. by functionally interacting with cell autonomous clues such as oncogenes (BRAF, KRAS), tumor suppressor genes (APC) and environmental clues such as MMPs.” Do you mean ‘ by promoting malignant progression initiated by cell autonomous clues such as oncogenes (BRAF, KRAS), tumor suppressor genes (APC) and environmental clues such as MMPs’??
“4. by the type of signaling pathway targeted by Rac1b. For instance, Rac1b promotes NFκB signaling but inhibits TGF- signaling.”Do you mean ‘ by the balance in the target tissue of signaling pathway targeted by Rac1b. For instance, Rac1b promotes NFκB signaling but inhibits TGF-b signaling’?
“5. by affecting physiological processes that predispose to cancer such as carcinogen/acute inflammation or protect against cancer such as early mucosal repair.” It is unclear how this is a level of eliciting the tumorigenic role. Do you mean ‘ by stimulating cancer-promoting processes such as carcinogen/acute inflammation or protecting against cancer such as early mucosal repair, depending on the tissue’ ?
22. paragraph 9.2 The final description of potential therapeutic intervention may allude to the possibility to develop protein-protein interaction inhibitors that could block RAC1b-specific downstream signalling.
Author Response
Reviewer 1
Comments and Suggestions for Authors
In this manuscript the authors pretended to provide a comprehensive overview on the biology of RAC1b in eukaryotic cells. Given the considerable increase in data published on this splice variant over the last 15 years, this pretension is fully justified.
The authors represent a rather compartmentalized overview, organized in subsections that separately describe gene data, physiological protein properties, biological processes, signaling pathways and finally cancer types. This necessarily leads to some degree of redundant information that is being referred to in various of the above mentioned subsections. However, this organization is useful for the review to be used as a look-up document regarding the roles of RAC1b in various biological aspects and allowed to gather the published information on these aspects that currently is scattered throughout the meanwhile quite numerous publications on RAC1b.
The manuscript contents are well-structured and represent well the state of the art. I have several mostly minor suggestions to improve the manuscript:
1. I recommend to add ‘epithelial-mesenchymal transition’ to the list of keywords.
Response: done
2. The authors may wish to follow the gene name nomenclature guidelines (http://www.genenames.org/about/guidelines#Appendix1) recommending that human gene symbols generally are italicised, with all letters in uppercase (e.g., RAC1), whereas the corresponding protein designations are the same as the gene symbol, but are not italicized (RAC1).
Response: We have followed these guidelines and have made the necessary changes throughout the manuscript.
3. paragraph 3, lines 50-59: It is stated that the RAC1 but not the RAC2 gene contains an additional exon; information on the RAC3 gene should be added, too.
Response: As requested, this piece of information has been added.
4. lines 51-53: The following sentence is ill-constructed: “The exon 3b of RAC1 51 contains additional 57 nucleotides between codons 75 and 76 of RAC1 and immediately behind the 52 switch II region”. The exon 3b itself contains neither codons 75 or 76 nor the switch 2 region. I suggest a fusion with the following phase: The exon 3b of RAC1 contains additional 57 nucleotides and this results in an in-frame insertion of 19 new amino acids between codons 75 and 76 of RAC1 immediately behind the switch II region, including two potential threonine phosphorylation sites for casein kinase II and protein kinase C.
Response: We agree with the reviewer and have replaced the sentence with the proposed version.
5. The last two sentences of this paragraph (lines 56-58) should be moved to the section 5.1 on cancer progression
Response: We followed the reviewers suggestion and have integrated these sentences in the first paragraph of section 5.1.
6. paragraph 4.1, line 70: The following sentence could be misleading “The insertion of exon 3b leads to enhanced intrinsic guanine nucleotide exchange activity and..”; this is not a real activity that RAC1b has; I suggest to refer only to nucleotide exchange and reduced affinity, for example ‘The insertion of exon 3b leads to a reduced affinity for GDP and consequently enhanced intrinsic guanine nucleotide exchange, as well as a decreased intrinsic GTPase activity, resulting in…
Response: We agree with the reviewer and have rephrased the sentence as suggested.
7. lines 73-81: references are missing here that showed the relation between RAC1b and Rho-GDI
Response: The relevant references have been added.
8. paragraph 4.2, line 86-94: SR protein family members of splicing factors have been renamed in 2010 (doi: 10.1101/gad.1934910), ASF/SF2 is now SRSF1 and SRp20 is SRSF3.
Response: We thank the reviewer for drawing our attention to this point. We have replaced the old names of both proteins with the new ones.
9. paragraph 5.1: refers to many aspects that are repeated below in other sections, I suggest to focus this paragraph on the two described transgenic mouse models.
Response: Despite the possibility of partial redundancy we think it is important to have these observations summarized here under the subheading of cancer progression/cellular transformation. The first paragraph also contains the last two sentences from section 3 (see response to #5).
10. paragraph 5.2: lines 197: the data described in the first line are not from Matos but Kotelevets and colleagues,
Response: This error has been corrected.
11. paragraph 5: Should mention the mechanistic relation of MMP3 with hnRNP A1 and Rac1b splicing given in ref 10
Response: This relationship has already been described in section 4.2., fourth paragraph.
12. paragraph 5.4: please include in this section the recently published paper on EMT in hepatocarcinoma cells (doi: 10.1186/s12964-018-0312-4) that is cited as reference 65, which is no longer in press and should be updated. I would also suggest to add a comment at the end of this section on the inconsistent observations that RAC1b can be an inducer or an inhibitor of EMT.
Response: This paper has been included in subsections 5.1., 5.4. and 5.6. The reference has been updated and a sentence on the inconsistent role of Rac1b has been added to the end of section 5.4. (this point is discussed in more detail in 8.1.).
13. paragraph 5.5: line 275, please indicate the cell type in which this was observed, in order to distinguish data clearly from those reported for keratinocytes in the same paragraph.
Response: The cell type (HT29 cells) has been added.
14. paragraph 5.6: Cell motility and migration would be a more appropriate subheading here
Response: The subheading has been changed as suggested.
15. paragraph 5.10, lines 410-418: The name Rac1b has been used in the past to describe a neuronal-specific chicken Rac gene (Malosio et al., 1997) that was later recognized as homologous to human Rac3 (Albertinazzi et al 2003) (corresponding to references 57 and 58 of the manuscript). The same applies to paragraph 6.6 on ER signalling and ref 64. The corresponding paragraphs should therefore be removed from the manuscript.
Response: We thank the reviewer for this important piece of information. As suggested, we have removed both paragraphs/subheadings.
16. paragraph 7: The described topics do not justify an independent section. The data on RAC1b interacting proteins (7.1) belong to the end of section 4, following topics on protein stability and subcellular localization.
Response: As recommended, we have moved subsection 7.1. to the end of section 4.
17. paragraph 7.2 on Inhibitors, should be moved to the end of section 6 on signaling pathways, as inhibitors are important tools for research or pharmacological intervention and their description fits better to this topic than to a section on protein interactions.
Response: As recommended, we have moved subsection 7.2. to the end of section 6.
18. Regarding the content of paragraph 7.2, both general RAC inhibitors (RAC1 and RAC1b), EHT1864 and NSC23766 should be discussed first before proceeding to the described RAC1b-specific inhibitor data. It would also be useful to distinguish that ibuprofen has been suggested as an inhibitor of the alternative splicing event rather than a direct inhibitor of RAC1b activity.
Response: As suggested, we followed the advice of the reviewer and have first discussed the general Rac1 inhibitors followed by the Rac1b-specific ones. The mechanistic basis of ibuprofen’s effect on Rac1b has been mentioned.
19. Finally, paragraph 7.3 on mutual negative regulation of RAC1 and RAC1b would fit much better to the end of section 5 that describes the roles on motility, EMT, cell survival etc, which could be modulated by the described mutual negative regulation. In line 555, ‘reduce’ should be ‘reduced’. At the end of line 558; I miss a phrase or expression suggesting that competition for RAC activation by GEFs as a likely mechanism (before proceeding to the remaining paragraph describing possible effects on protein stability).
Response: Subsection 7.3. has been moved to the end of section 5. A sentence on possible competition for Rac activation by GEFs as a likely mechanism for mutual negative regulation has been added (page 12, lines 468-469).
20. paragraph 8.2: The described association of RAC1b overexpression with the presence of BRAF-V600E should be discussed with respect to the recently proposed Consensus Molecular subtype CMS1 of colorectal cancer (doi:10.1038/nm.3967).
Response: As suggested, a brief discussion has been added to the new subsection 7.2. (lines 595-603).
21. paragraph 9.1: This is a very useful and important section that integrates conflicting data form the literature; however, the following paragraph (lines 639-650) needs some clarification considering the given topic” RAC1b’s tumorigenic role can be elicited at several levels:”:
“1. by activity regulation based on the biochemical characteristics.” is unclear. Do you mean ‘ by an increase in RAC1b activation levels due to altered regulation of GEFs and GAPs’??
Response: Yes, this is essentially what we wanted to say.
“3. by functionally interacting with cell autonomous clues such as oncogenes (BRAF, KRAS), tumor suppressor genes (APC) and environmental clues such as MMPs.” Do you mean ‘ by promoting malignant progression initiated by cell autonomous clues such as oncogenes (BRAF, KRAS), tumor suppressor genes (APC) and environmental clues such as MMPs’??
Response: Yes, this is essentially what we wanted to say. However, we would like to leave the term “functional interactions” here.
“4. by the type of signaling pathway targeted by Rac1b. For instance, Rac1b promotes NFκB signaling but inhibits TGF-b signaling.”Do you mean ‘ by the balance in the target tissue of signaling pathway targeted by Rac1b. For instance, Rac1b promotes NFκB signaling but inhibits TGF-b signaling’?
Response: Yes, using the term “balance” here is a good suggestion. We have rephrased this sentence accordingly.
“5. by affecting physiological processes that predispose to cancer such as carcinogen/acute inflammation or protect against cancer such as early mucosal repair.” It is unclear how this is a level of eliciting the tumorigenic role. Do you mean ‘ by stimulating cancer-promoting processes such as carcinogen/acute inflammation or protecting against cancer such as early mucosal repair, depending on the tissue’ ?
Response: Yes, the phrasing suggested by the reviewer is much more precise.
Additional changes made: 1. We have changed the order of the various points, 2. Following the list of Rac1b’s tumorigenic role, we have added, for the sake of completion, two observations that are compatible with an anti-tumorigenic role.
22. paragraph 9.2 The final description of potential therapeutic intervention may allude to the possibility to develop protein-protein interaction inhibitors that could block RAC1b-specific downstream signalling.
Response: As suggested, a sentence expressing this possibility has been added at the end of this section (lines 750-751).
Reviewer 2 Report
In this review article, entitled “Rac1b, a Rho GTPase with versatile functions in malignant transformation and tumor progression”, the authors tried to comprehensively describe the current knowledge of Rac1b biology. The manuscript is concise and well written. However, especially in the section 5 “Biological Function of Rac1b”, the authors listed so much information very concisely without any visual aid, leading to difficulties in readily understand the contents. Adding summary figures for long and dense subsections such as “5.4 EMT” in the section 5 would make this article more readable for most of the readers.
Other points:
1) It is better to always give full forms (such as “epithelial-mesenchymal transition”, “colorectal cancer”) along with abbreviations (“EMT”, “CRC”) in the section/subsection title.
2) The sentence at L246-7, "In contrast, … by fibronectcin [16].” appears grammatically incorrect.
3) At L254-8, an adequate citation that reported regulation of E-cadherin transcription by Rac1b appears missing.
Author Response
Reviewer 2
Comments and Suggestions for Authors
In this review article, entitled “Rac1b, a Rho GTPase with versatile functions in malignant transformation and tumor progression”, the authors tried to comprehensively describe the current knowledge of Rac1b biology. The manuscript is concise and well written. However, especially in the section 5 “Biological Function of Rac1b”, the authors listed so much information very concisely without any visual aid, leading to difficulties in readily understand the contents. Adding summary figures for long and dense subsections such as “5.4 EMT” in the section 5 would make this article more readable for most of the readers.
Response: As suggested, we have included two more figures in sections 5 and 6. Figure 2 has been moved to subsection 5.4. on EMT.
Other points:
1) It is better to always give full forms (such as “epithelial-mesenchymal transition”, “colorectal cancer”) along with abbreviations (“EMT”, “CRC”) in the section/subsection title.
Response: This has been rectified (marked in red color).
2) The sentence at L246-7, "In contrast, … by fibronectcin [16].” appears grammatically incorrect.
Response: We have removed “by” from this sentence and it now reads correctly.
3) At L254-8, an adequate citation that reported regulation of E-cadherin transcription by Rac1b appears missing.
Response: The appropriate citation (#20) has been added.
Reviewer 3 Report
In this review manuscript, the authors have provided an extensive and up-to-date overview of the small GTPase Rac1b, a splice variant of the canonical Rac1. The review is very useful, because of its comprehensive nature and currency.
My most significant concern is that, in attempting to be comprehensive, the authors have not been sufficiently critical of some of the findings in the literature. There are several examples where there are contradictory findings regarding the function or activity of Rac1b, which is explained as possibly being due to cell type differences. It is also possible that the contradictions are due to study design/implementation, and that rigorous critical reading would help to resolve some of these contradictions. This would add considerable value to the review.
Similarly, the authors have been careful to question the selectivity of the reported Rac1 inhibitor sanguinarine, which makes its inclusion in the manuscript questionable. If the selectivity is not convincing, then the findings that have been made using the compound are not robust and not worthy of inclusion.
Given the similarity of the effect of the Rac1b insertion on increasing the rate of spontaneous GTP/GDP exchange to the effect of the Rac1 P29S mutation commonly observed to co-occur with BRAF or NRAS mutations, it would be useful to point out the similarities and differences.
This comparison between Rac1b and Rac1 P29S, both being fast-cycling versions, would also be worth including when discussing the differences in findings between Rac1b and activated versions of Rac1 (e.g. Q61L). There are several instances in the review that indicate studies have used the different activities of Rac1b vs activated Rac1 to argue that there are differences in the activities of the two splice variants. An additional possibility that should be highlighted is the differences between fast-cycling proteins vs GTP-locked mutants. It may well be that the act of cycling is a key event, rather than just being in the active state constitutively.
There is a constant switching in referencing style, between referring to specific studies by author name(s) or simply referring to the findings without mentioning authors. It would be better to use a single style, one or the other. This becomes particularly confusing in the paragraph between lines 475-487, where no author names are mentioned, but references are made to "this group", "these authors", and "the same group". The manuscript should be made more consistent throughout.
Additional specific comments:
Lines 29-36 - this introduction is not well focused on Rac1b, instead it's a justification for why the reader should not dismiss the manuscript as just another review about Rac1. It would be more helpful if the introduction were to indicate the areas that will be covered in the rest of the manuscript.
Line 43 - which Rac?
Lines 131-132 - it's not clear at this point in the review how ubiquitinylation of Rac1 occurring through a JNK activated process explains the defective ubiquitinylation of Rac1b. Please explain.
Lines 252-254 - I'm not convinced that ROS is mandatory for EMT, stemness and dissemination of metastatic cells. This seems like an over-statement and should be re-written.
Lines 317-320 - as discussed above, the findings based on the use of the non-selective compound sanguinarine are not convincingly indicative of a function of Rac proteins.
Line 337 - promitotic isn't the correct word in this context, its proliferation in general that is being referred to rather than mitosis.
Lines 467-468 - see comment 4 above.
Finally, I would recommend that the authors make sure that abbreviations are explained at their first occurrence. I'm not sure that OIS is explained at all, others are indicated well after they make their first appearance.
Author Response
Reviewer 3
Comments and Suggestions for Authors
In this review manuscript, the authors have provided an extensive and up-to-date overview of the small GTPase Rac1b, a splice variant of the canonical Rac1. The review is very useful, because of its comprehensive nature and currency.
My most significant concern is that, in attempting to be comprehensive, the authors have not been sufficiently critical of some of the findings in the literature. There are several examples where there are contradictory findings regarding the function or activity of Rac1b, which is explained as possibly being due to cell type differences. It is also possible that the contradictions are due to study design/implementation, and that rigorous critical reading would help to resolve some of these contradictions. This would add considerable value to the review.
Response: As requested, we have included the possibility that the contradictions are due to study design/implementation (see 8.1.). The contradictions emerged from listing the main results taken from the summaries of articles published in PubMed. Only those findings that were clear and were from established and recognized groups were considered and we therefore believe that further critical reading would not help resolving these differences. However, in one paragraph (line 378-379) we have rephrased the sentence to make clear that the observation that Rac1b did not stimulate transcription from the cyclin D1 promoter does not necessarily mean that Rac1b does not promote proliferation.
Similarly, the authors have been careful to question the selectivity of the reported Rac1 inhibitor sanguinarine, which makes its inclusion in the manuscript questionable. If the selectivity is not convincing, then the findings that have been made using the compound are not robust and not worthy of inclusion.
Response: Since the selectivity of sanguinarine is highly questionable, we have removed the findings and the reference (#41) on this compound from the revised version.
Given the similarity of the effect of the Rac1b insertion on increasing the rate of spontaneous GTP/GDP exchange to the effect of the Rac1 P29S mutation commonly observed to co-occur with BRAF or NRAS mutations, it would be useful to point out the similarities and differences.
Response: As suggested, we have briefly discussed similarities and differences of Rac1-P29S and Rac1b (page 2, lines 73-78).
This comparison between Rac1b and Rac1 P29S, both being fast-cycling versions, would also be worth including when discussing the differences in findings between Rac1b and activated versions of Rac1 (e.g. Q61L). There are several instances in the review that indicate studies have used the different activities of Rac1b vs activated Rac1 to argue that there are differences in the activities of the two splice variants. An additional possibility that should be highlighted is the differences between fast-cycling proteins vs GTP-locked mutants. It may well be that the act of cycling is a key event, rather than just being in the active state constitutively.
Response: The idea of using fast cycling mutants of Rac1 rather than GTP-locked ones is very appealing. A paragraph dealing with this issue has therefore been added to section 8.1. (marked in red color, lines 707-713).
There is a constant switching in referencing style, between referring to specific studies by author name(s) or simply referring to the findings without mentioning authors. It would be better to use a single style, one or the other. This becomes particularly confusing in the paragraph between lines 475-487, where no author names are mentioned, but references are made to "this group", "these authors", and "the same group". The manuscript should be made more consistent throughout.
Response: As requested, we have made the manuscript more consistent by referring to the relevant studies by number.
Additional specific comments:
Lines 29-36 - this introduction is not well focused on Rac1b, instead it's a justification for why the reader should not dismiss the manuscript as just another review about Rac1. It would be more helpful if the introduction were to indicate the areas that will be covered in the rest of the manuscript.
Response: We have deleted the sentence with the justification and have added the areas that are covered in this manuscript.
Line 43 - which Rac?
Response: The term “Rac” refers to a common ancestor of Rac1,-2, and -3.
Lines 131-132 - it's not clear at this point in the review how ubiquitinylation of Rac1 occurring through a JNK activated process explains the defective ubiquitinylation of Rac1b. Please explain.
Response: This scenario has been explained. A missing piece of information here is the observation that Rac1b cannot activate JNK which in turn is required for ubiquitination (page 4, lines 134-138).
Lines 252-254 - I'm not convinced that ROS is mandatory for EMT, stemness and dissemination of metastatic cells. This seems like an over-statement and should be re-written.
Response: The subsentence “which is mandatory for…cells” has been removed.
Lines 317-320 - as discussed above, the findings based on the use of the non-selective compound sanguinarine are not convincingly indicative of a function of Rac proteins.
Response: The findings based on sanguinarine and the corresponding citation (#41) have been removed from the revised version (see also above).
Line 337 - promitotic isn't the correct word in this context, its proliferation in general that is being referred to rather than mitosis.
Response: As requested, we have replaced “promitotic” by proliferation-promoting”.
Lines 467-468 - see comment 4 above.”
Response: The subsentence “which is mandatory for…cells” has been removed.
Finally, I would recommend that the authors make sure that abbreviations are explained at their first occurrence. I'm not sure that OIS is explained at all, others are indicated well after they make their first appearance.
Response: OIS is explained at first mention in line 180 of the revised version. Other abbreviations are now explained at their first appearance.
Round 2
Reviewer 1 Report
The Authors have adequately addressed the concerns and suggestions